# Revealing the Evolution from Q-Switching to Mode-Locking in an Erbium-Doped Fiber Laser Using Tungsten Trioxide Saturable Absorber

**Xin Tan [1], Ya Liu [1,\*], Yongkang Zheng [1], Zewu Xie [1] and Guoqing Hu [2,3]**

1 Yunnan Key Laboratory of Opto-Electronic Information Technology, School of Physics and Electronic Information Technology, Yunnan Normal University, Kunming 650500, China
2 Key Laboratory of the Ministry of Education for Optoelectronic Measurement Technology and Instrument, Beijing Information Science & Technology University, Beijing 100192, China
3 Beijing Laboratory of Optical Fiber Sensing and System, Beijing Information Science & Technology University, Beijing 100016, China
\* Correspondence: liuya@buaa.edu.cn

**Abstract:** Passively Q-switching and mode-locking technologies can generate short pulses with durations that differ by several orders of magnitude widely used in different applications. Recently, Q-switching and mode-locking realized in an identical laser cavity with saturable absorbers was reported. The analysis of pulse conversion is helpful for us to further understand the pulse dynamics of a laser. In this paper, the pulse evolution from Q-switching, Q-switched mode-locking to mode-locking, is demonstrated by using a tungsten trioxide saturable absorber in a ring-cavity erbium-doped fiber laser. Firstly, self-started Q-switching at 1563 nm is observed, the repetition rate continuously increases, and the duration decreases when the pump power increased. Then, with an adjusting intra-cavity state of polarization under a high pump power level, stable Q-switched mode-locking pulses evolved from Q-switching, are observed. The amplitude of the emerged pulse sequence with a period of 36.8 ns, determined by cavity length, is modulated by the Q-switched envelope with the period of 10.3 μs. By optimizing the intracavity polarization carefully, stable continuous wave mode-locking operation is achieved eventually. To the best of our knowledge, this is the first experimental demonstration of Q-switching and mode-locking, respectively, in an identical transition-metal-oxides-based pulsed fiber laser without modification of cavity structure.

**Keywords:** pulsed fiber lasers; Q-switching; mode-locking; tungsten trioxide

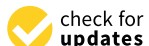



## 1. Introduction

Short-pulse lasers have shown high potential in applications regarding nonlinear optics and ultrafast optics, such as communications, optical sensing, and laser processing [1,2]. Passively Q-switching and mode-locking are typical technologies for generating short and ultrashort pulse laser in fiber lasers [3,4]. The Q-switching (QS) is a technology for modulating the laser cavity quality factor (Q). By changing the Q value of the optical resonator, the energy stored in the activated medium is released instantaneously. Therefore, the pulses formed by this mechanism usually have a duration in the order of microseconds and a repetition rate of kilohertz. Meanwhile, mode-locking (ML) is achieved by introducing a fixed phase relationship between the modes of the laser cavity. Compared with QS technology, ML technology produces a shorter pulse duration and higher peak power, but a lower pulse energy. The different characteristics make them suitable for diverse applications.

In the past decade, saturable absorbers (SA) have been increasingly employed in fiber laser cavities for passively Q-switched or mode-locked operation to generate ultrashort pulses, due to their excellent nonlinear absorption optical properties. Moreover, the same SA sample could achieve pulse transition from QS to ML by using two common methods.

One is introducing the nonlinear polarization evolution (NPE) effect to the fiber laser by using a polarization-dependent component, such as a polarizer [5], tapered fiber [6–8], or D-shaped fiber [9]. The other is dispersion management with significantly increased cavity length [10–20]; however, the repetition frequency of the laser is greatly reduced. In addition to the dispersion and nonlinearity, the parameters of the SA can also affect the performance of the QS and ML laser. When the SA have a lower saturation intensity and thermal damage resistance, transition from QS to ML can also be achieved when the pump power is exponentially higher than the saturation intensity. Among the various materials for fabricating the SA, nanomaterial saturable absorbers have been widely investigated due to their wide operating bandwidth, low cost, and easy integration. Moreover, recent studies have found that transition metal materials have excellent optical properties ultrafast response time and broadband absorption [21–23]. Especially, the transition metal tungsten oxide shows excellent performance to fabricate SA for its good air stability, thermal damage thresholds, and third-order nonlinearity [24]. In 2020, Al-Hiti's team used $WO_3$ as SA for the first time to achieve QS pulse output at 1556.8 nm. Additionally, in the same year, based on the $WO_3$-SA with a saturation intensity of 0.04 $MW/cm^2$, a mode-locked erbium-doped fiber laser (EDFL) was obtained by introducing a segment single-mode fiber (SMF) with a length of 100 m in the cavity. However, the repetition frequency is as low as 1.85 MHz [25].

In this paper, we deposited $WO_3$ nanometer material on the fiber end face by using optical driven deposition method to fabricate a fiber-integrated SA with a low saturation intensity. In the experiment, a complete pulse evolution from QS, QML to ML was observed by increasing the pump power. In the whole process, the structure of the laser cavity is kept unchanged. The total cavity length is 7.5 m, and the repetition rate of obtained mode-locked pulse is 27.2 MHz. For the QS state, the pulse energy undergoes an increase first and then a rapid decrease with monotonically increased pump power. At high pump levels, the QS envelope becomes rough. In this case, mode-locked pulse sequence would emerge with significantly amplitude-modulated by the QS envelope, i.e., QML operation, when the polarization state is adjusted. Moreover, the QML could eventually evolved into ML for an optimizing polarization state. This work achieves an evolution from QS to ML without modification of fiber cavity and it would contribute to a better understanding of the establishment of mode-locking.

## 2. $WO_3$-SA Preparation and Characterization

The setup for depositing $WO_3$ nanoparticles on a clean optical fiber ferrule connector [26,27] is shown in Figure 1a. $WO_3$ is an alcohol-soluble material. Firstly, the $WO_3$ solution was prepared by mixing approximately 0.5 mg of $WO_3$ powder with 2 mL of ethanol and ultrasonicated for 20 min. Next, a single-mode fiber (SMF) end was immersed in the solution and injected with 980 nm LD laser at an empirical power of 25 mW for 5–10 min. The clean and the deposited fiber end-face are shown in Figures 1b and 1c, respectively, obtained by taking photos of an optical fiber end-face inspection microscope. Finally, the fiber ferrule connector with fiber end-face deposited $WO_3$ nanoparticles layer was connected with another clean fiber connector through an adapter to compose a $WO_3$-based SA, as shown in Figure 1d.

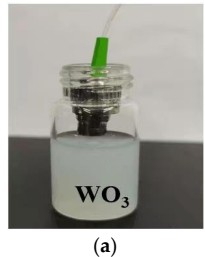 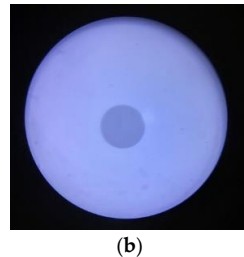 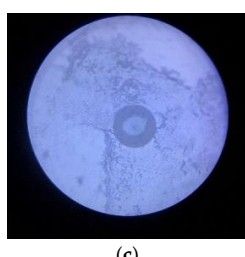 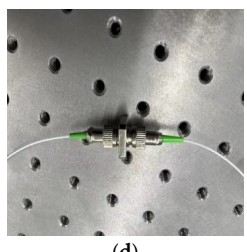

(a)     (b)     (c)     (d)

**Figure 1.** (**a**) The setup for $WO_3$ deposition; (**b**,**c**) the fiber end-face before and after deposited; and (**d**) photograph of the fiber-integrated $WO_3$ saturable absorber.

Figure 2a shows the morphological characteristics of the WO$_3$ solution sample used for deposition by using field emission scanning electron microscopy (SEM), which shows the morphology of the WO$_3$ as spherical nanoparticles. It can be clearly observed that the WO$_3$ is uniformly distributed in a spherical nanoparticle structure.

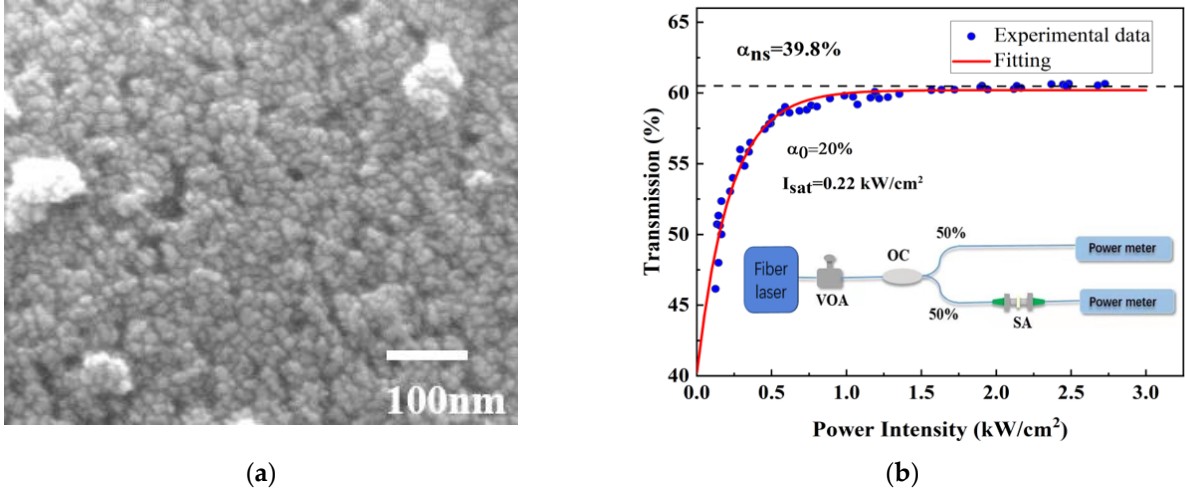

**(a)**　　　　　　　　　　　　　　　　　　　**(b)**

**Figure 2.** (**a**) SEM image of WO$_3$ nanomaterials. (**b**) The nonlinear absorption of the WO$_3$-SA.

In addition, we investigated the optical nonlinear absorption properties of the fabricated WO$_3$-SA based on the twin-detector measurement approach, as shown in the inset of Figure 2b. The probe light source is a homemade passively mode-locked fiber laser at 1550 nm with a pulse duration of 0.5 ps and a fundamental frequency of 24.4 MHz. The nonlinear transmission curve and the fitting curve are shown in Figure 2b. The experimental data are fitted by the Formula (1):

$$\mathrm{T}(I) = 1 - \alpha_0 exp\left(\frac{-I}{I_{sat}}\right) - \alpha_{ns} \tag{1}$$

where $\mathrm{T}(I)$ is transmission coefficient, $\alpha_0$ is modulation depth, $I$ is input intensity, $I_{sat}$ is saturation intensity, and $\alpha_{ns}$ is non-saturation loss. According to the fitting data, the nonlinear absorption characteristic is illustrated with a non-saturation loss of 39.8%, a large modulation depth of 20% and a low saturation intensity of 0.22 kW/cm$^2$. The parameters are similar to that of SA for self-Q-switching as in Ref. [28].

### 3. Experimental Setup

The schematic of the fiber laser configuration is shown in Figure 3. A piece of 2.5-m-long erbium-doped fiber (EDF: YOFC, EDF22/6/125-23) with 20 dB/m absorption coefficient at 1529 nm is used as the gain medium, forward-pumped by a 980 nm pump laser. A polarization-independent isolator (PI-ISO) is used to assure unidirectional lasing. A polarization controller (PC) is used to adjust the intra-cavity polarization states. The nonlinear parameters of the WO$_3$-SA used in the cavity are described in the previous section, and its linear insertion loss is 1.2 dB. A 90/10 fiber coupler is used to extract 10% as the output. The total length of the cavity is about 7.6 m, and all the fiber pigtails are single-mode fiber (SMF). The dispersion of the EDF is estimated as −6 ps/km/nm at 1560 nm, and the total net cavity dispersion is about −0.09 ps$^2$. To investigate the polarization-dependent characteristics of the fiber cavity, we measured the output power of the laser emitting continuous wave (CW) under different polarization states by tuning the PC. Under a same pump level, a maximum fluctuation of 10% for the output power is observed, indicating the polarization-dependent loss (PDL) is measured less than 0.5 dB.

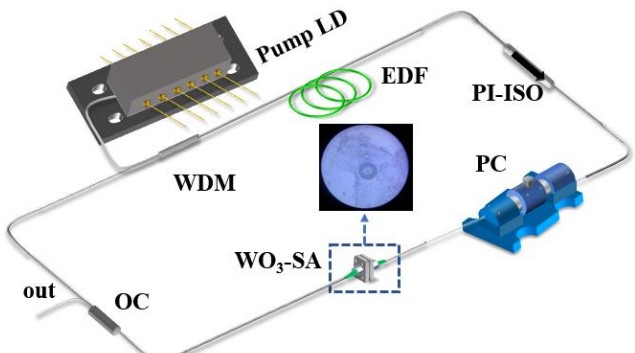

**Figure 3.** Schematic of the erbium-doped fiber ring laser with $WO_3$-SA.

In this experiment, the output of the fiber laser is characterized by an optical spectrum analyzer (Yokogawa, AQ6370C), a 1 GHz photodetector (Newport, 1611FC-AC), a digital oscilloscope (Tektronix, TDS3054C) with a 500 MHz bandwidth and a sampling rate of 5 GS s$^{-1}$, and a radio frequency (RF) spectrum analyzer (Agilent, E4405B).

## 4. Results and Discussions

### 4.1. Q-Switched Operation

The self-started Q-switching is emerged at 26 mW of 980 nm pump power. The optical spectrum of QS is shown in Figure 4a, with a center wavelength of 1562.5 nm and a 3-dB bandwidth of 1.4 nm. Compared with the spectrum that without SA inserted in the fiber laser, the wavelength is slightly blue shifting due to the additional insertion loss of the SA. The corresponding QS average output power is 1.1 mW. The pulse duration, period, repetition rate and single pulse energy are estimated to be 8.2 μs, 29.6 μs, 33.8 kHz and 33 nJ, respectively.

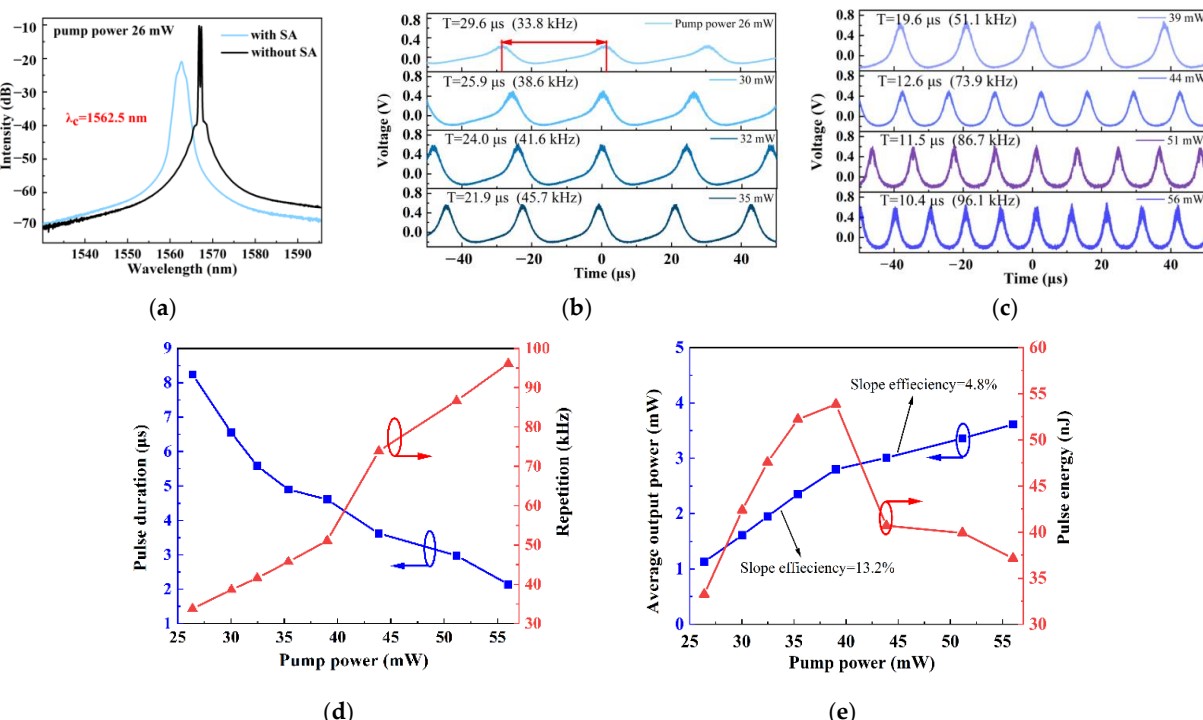

**Figure 4.** Laser performance of $WO_3$-based passively Q-switched EDFL. (**a**) Output optical spectrum. (**b**,**c**) Oscilloscope traces at different pump power. (**d**) Pulse duration and repetition rate versus pump power. (**e**) Average output power and pulse energy versus pump power.

The laser pulse trains at the different pump power level from 26 mW to 56 mW are shown as Figure 4b,c, respectively. Both the repetition rate and pulse duration are pump power dependent, which is a typical behavior of a Q-switched laser [29]. As the pump power gradually increases from 26 mW to 56 mW, the repetition rate increases from 33.8 kHz to 96.1 kHz, the pulse duration decreases from 8.2 μs to 2.1 μs, the average output power increases from 1.1 mW to 3.6 mW, and the corresponding pulse energy fluctuates from 33.2 nJ to 53.9 nJ. The measured data are summarized as shown in Figure 4d,e. Obviously, when the pump power is set above 39 mW, the average output power increases slowly while the corresponding pulse energy of the QS pulse decreases rapidly. The estimated slope efficiency of the laser is not constant but drops from 13.2% to 4.8%. Meanwhile, the observed amplitude on the oscilloscope shown in Figure 4b,c do not increase significantly anymore and the trace becomes rough. This phenomenon may attribute to the thermal accumulation and supersaturation of SA at high pumping level [7,20,30]. At QS stage, the pulse intensity is greatly improved due to the bleaching of the saturable absorber and the reduced intra-cavity loss. However, because of the high pulse intensity, the gain saturation effect is induced, and the population inversion in the laser is greatly consumed, which leads to the sharp weakening of the gain in the cavity. Under this condition, it is impossible to maintain such high-power pulse operation, resulting in the Q-switching instability, which is manifested as a rapid decrease of the intensity and bandwidth of the spectrum.

### 4.2. Q-Switched Mode-Locked Operation

To investigate the pulse evolution, we continuously adjusted the polarization state of the cavity through rotating the intra-cavity PC, when keeping the pump power at 56 mW. A complete transition process is observed from Q-switching to Q-switched mode-locking, eventually continuous wave (CW) mode-locking (ML) state. The waveforms of different operation states are shown in Figure 5a–e. In the process of fine-tuning of PC, the rough QS pulse envelope gradually reveals that it is consist of a large quantity of narrow pulses modulated by inconsistent amplitude, i.e., Q-switched mode-locking (QML). The QML is commonly interpreted as Q-switching instability in a passively mode-locked laser. This is a transition state of Q-switched and mode-locked regime [31], which could be achieved from QS pulse or evolve to ML pulse by utilizing additional nonlinear saturable absorption. However, at a certain condition, the gain, loss, dispersion, saturable absorption, and other nonlinear effects in the laser reach a balance state, the stable QML operation could be realized [32]. We attribute the evolution in this paper to the additional weak nonlinear saturable absorption introduced by the indeed weak polarization sensitivity of the laser when the PC adjusted, as mentioned in the section of experimental setup. The details on the recorded pulse traces in spans of 1 μs for the operation states in Figure 5a–e are shown in Figure 5f–j, respectively, which identify the establishing of mode-locking. In this condition, the average output power was measured of 3.7 mW, without significantly variation compared to that of the QS state. The pulse period is measured about 36.8 ns and the corresponding repetition frequency is 27.2 MHz, in good agreement with the total cavity length. Additionally, the waveform observed is close to sinusoidal with a pulse width of about 17 ns for the state 3 of QML.

### 4.3. Mode-Locked Operation

With the further optimization of intra-cavity state of polarization and keeping the same pump power of 56 mW, the mode-locked pulse duration in the ML state could be changed continuously. In the experiment, the pulse narrowing process can be observed from the oscilloscope by carefully adjusting the PC. Figure 6a,b and the insets show two narrowed pulse trains with durations of 9.0 ns to 2.1 ns, respectively. The pulse widths are estimated by the oscilloscope with a bandwidth of 500 MHz and a sampling rate of 5.0 GS s$^{-1}$ followed a photodetector with 1 GHz bandwidth and 400 ps rise time. The negative overshoot of the signal is aroused by the response characteristics of the detector preamplifier [3]. We could find that there is barely intensity modulation over a 2-μs span, which indicates

the high stability of continuous wave mode-locking operation. This evolution process is also attributed to the additional nonlinear saturable absorption introduced by the weak polarization sensitivity of the laser.

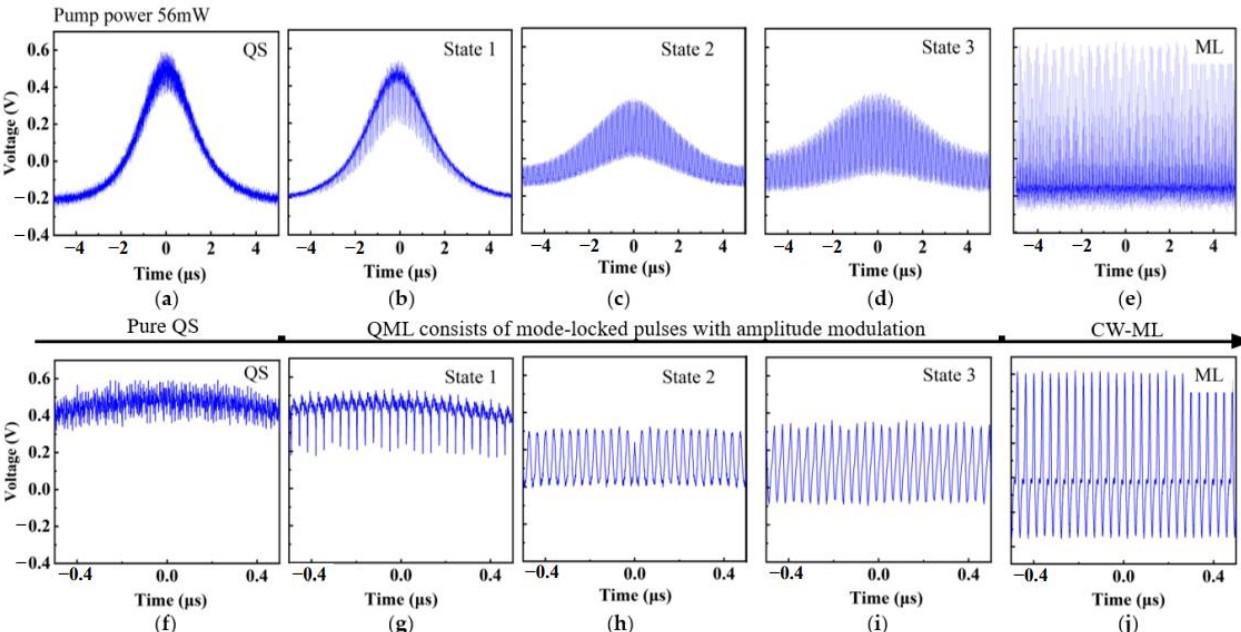

**Figure 5.** The transition of pulsed operation conversion of the $WO_3$-SA laser at the 56 mW pump power. (**a**) Pure QS; (**b**–**d**) QML; (**e**) CW ML and (**f**–**j**) Details of the corresponding pulse traces recorded by oscilloscope.

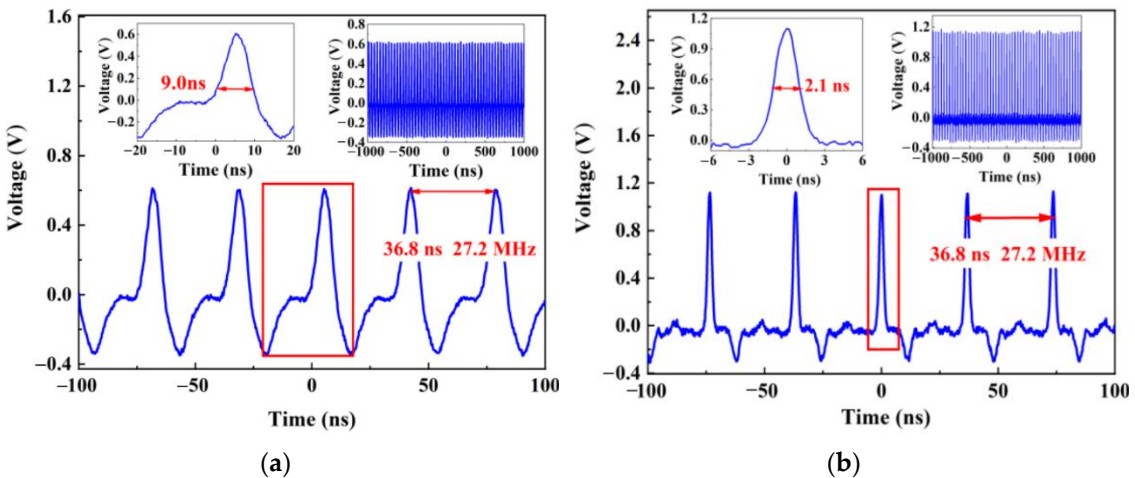

**Figure 6.** The mode-locked pulse trains under different polarization states at pump power of 56 mW. (**a**) Narrowed pulses with durations of 9.0 ns. (**b**) Narrowed pulses with durations of 2.1 ns.

In addition, the optical spectrum and RF spectrum are measured for the pulse with duration of 2.1 ns, as shown in Figures 7a and 7b, respectively. The optical spectrum is fitted with a $sech^2$ function and a 3-dB spectral bandwidth of 1.5 nm is estimated. A SNR of 46.7 dB at 27.2 MHz on the RF spectrum indicating a continuous wave mode-locking operation is achieved. In this case, the average output power of the ML laser is 3.9 mW, larger than that under QML or QS operation at a same pump power. We attribute this phenomenon to the high transmittance of saturable absorption for high peak power.

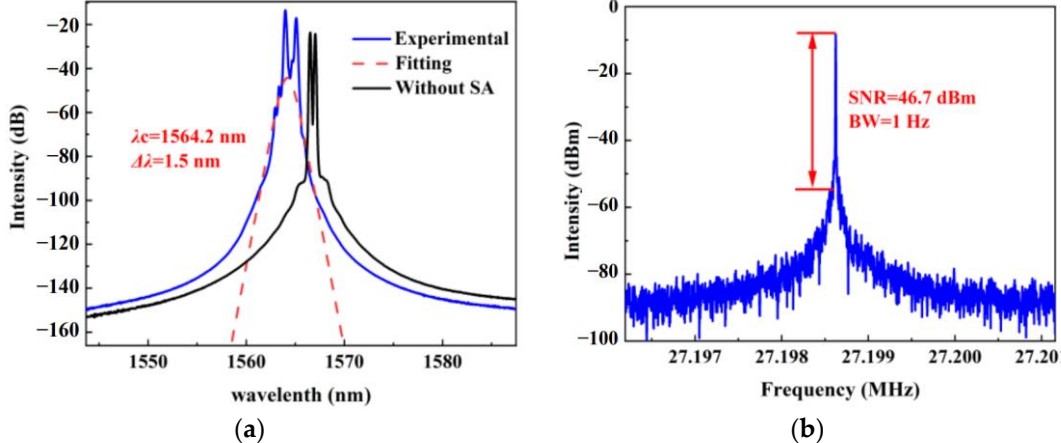

**Figure 7.** The characteristics of the mode-locked laser with 2.1 ns pulse duration. (**a**) Optical spectrum. (**b**) RF spectrum.

Finally, we summarize the performance of the pulse transition from QS to ML in EDFL based on increasing the cavity length to balance the dispersion and nonlinearity in fiber cavity, see Table 1.

**Table 1.** Performance comparison for the work of SA to achieve pulse transition from QS to ML by balancing dispersion and nonlinearity in the EDFL.

| Year | SA Material | Saturation Intensity | Q-$\tau$ [μs] | QS-RF [kHz] | ML-RF [MHz] | Increased Length (SMF) | Net Cavity Dispersion | Ref. |
|------|-------------|----------------------|---------------|-------------|-------------|------------------------|------------------------|------|
| 2016 | $MoS_2$ film | / | 5.18–3.53 | 72.74–86.39 | 5.78 | 30 m | / | [17] |
| 2018 | $ReS_2$ film | 74 MW/cm$^2$ | 23–5.5 | 12.6–19 | 5.48 | 30 m | −0.57 ps$^2$ | [10] |
| 2019 | $Ho_2O_3$ film | 140 MW/cm$^2$ | 0.64 | 115.8 | 17.1 | 3 m | −0.19 ps$^2$ | [18] |
| 2020 | $TiS_2$ | 19.97 MW/cm$^2$ | 2.34 | 13.17–48.45 | 3.43 | 50 m | / | [12] |
| 2020 | MEH-PPV film | 40 MW/cm$^2$ | 3.54 | 59.5–78.6 | 1.86 | 100 m | / | [20] |
| 2020 | Te film | / | 8.9–5.2 | 15.9–47.6 | 5.04 | 40 m | / | [14] |
| 2021 | AZO film | 1437 MW/cm$^2$ | 2.2 | 86 | 1.86 | 100 m | / | [16] |
| 2021 | $Ti_3C_2T_x$ film | 20 MW/cm$^2$ | 24.53–13.07 | 40.75–76.48 | 1.89 | 100 m | | [19] |
| 2022 | $WO_3$ | 0.22 kW/cm$^2$ | 8.2–2.1 | 33.8–96.1 | 27.2 | 0 m | −0.09 ps$^2$ | This work |

## 5. Conclusions

We demonstrate a pulsed erbium-doped fiber laser based on $WO_3$, which can operate at Q-switching and mode-locking, respectively. The transition of the regimes from QS to QML and then ML is obtained and investigated by increasing the pump power and carefully adjusting the polarization state. Under a pump power of 26 mW, the QS are self-started and generated pulses with a duration of 8.2 μs and a repetition rate of 33.8 kHz. As the pump power increased, the repetition rate is increased significantly and then the oscilloscope trace becomes rough. When keeping at a high pump level, QML with series of mode-locked pulses (27.2 MHz) modulated by QS envelope (97.2 kHz) is obtained by adjusting the intra-cavity polarization state. Finally, by optimizing the polarization state, a stable continuous wave ML can be obtained with a minimal pulse duration of 2.1 ns. The results verify that QS and ML could be generated by an identical $WO_3$-based fiber laser. In addition, the QML as a transition state for QS and ML could be observed, which is helpful for the comprehension of the pulse formation mechanism in $WO_3$-based fiber laser.

**Author Contributions:** X.T. and Y.L. designed the experiments, characterized the saturable absorber, analyzed the data, and drafted the manuscript; Y.Z. and Z.X. provided experimental assistance; G.H. performed the theoretical analysis. All authors participated in the editing of the manuscript. All authors have read and agreed to the published version of the manuscript.

**Funding:** This work was financially supported by National Natural Science Foundation of China (NSFC) (61705236, 62105038), R&D Program of Beijing Municipal Education Commission (KM202211232001).

**Conflicts of Interest:** The authors declare that they have no competing financial interests or personal relationships that could have appeared to influence the work reported in this paper.

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
