# Peer review of "Revealing the Evolution from Q-Switching to Mode-Locking in an Erbium-Doped Fiber Laser Using Tungsten Trioxide Saturable Absorber"

_photonics, doi:10.3390/photonics9120962_

Round 1
Reviewer 1 Report (Previous Reviewer 1)
I am satisfied how the authors have thoroughly revised their work to address the issues raised in my previous report and to considerably improve the value and readability of the paper.
In view of this improvement I recommend the publication of the paper.
Reviewer 2 Report (Previous Reviewer 2)
In view of the authors' accurate answers to the questions raised by the reviewers, I suggest that the editors accept and publish the manuscript.
Reviewer 3 Report (Previous Reviewer 3)
Dear authors,
Thank you for your reply, my questions are solved. I think your paper now is good enough to publish.
Regards
This manuscript is a resubmission of an earlier submission. The following is a list of the peer review reports and author responses from that submission.
Round 1
Reviewer 1 Report
The authors present a manuscript based in an erbium doped fiber laser using WO3 as saturable absorber. The paper shows three modes of operation of the laser: Q- switch (Q-S), Q-switch mode locking ( QML) and mode locking ( ML).
First of all I want to point out that build a laser as the one presented here is a great achievement. This experimental work implies a very large amount of dedication and specifics skills.
However, this laser is not original. For example, Al - Hiti et al. present, two years ago, the same cavity, with the same active medium ( erbium doped fiber) and the same saturable absorber (Tungsten trioxide (WO3) film absorber for generating soliton mode-locked pulses in erbium laser, Optics and Laser Technology, 131 ( 2020) 106429) with even better performance that this manuscript.
One major claim of the paper is that " this is the first experimental demonstration of a transition-metal-oxides-based pulsed laser which can operate at three states of Q-switching, Q-switched mode-locking and mode-locking." This is not surprising, it is well kwown that the use of a saturable absorber as a passive mode locker can introduce a tendency for Q-switched mode-locked operation. Q-switched mode locking is an operation regime of a passively mode-locked laser where the intracavity pulse energy undergoes large oscillations, related to a dynamic instability (called Q-switching instability) related to undamped relaxation oscillations. The origin of Q-switching instabilities is that the saturable absorber typically “rewards” higher pulse energies with lower resonator losses, so that the damping of the relaxation oscillations is reduced. Besides, in general QML is regarded as a negative condition that we want to remove.
For this reason I looked with great interest the sentence " Q-switched mode-locked lasers have higher peak powers [23,24] making them uniquely attractive in certain applications. I read in ref. 24 : "... Q-switched mode-locked lasers are attractive in some applications, such as precision structure fabrication, nonlinear frequency conversion and medical equipment ..." Following the reference on that paper I finally find "Non-thermal ablation of neural tissue with femtosecond laser pulses", Appl. Phys. B 66, 121–128 (1998), which present a classical scheme where a mode locked Ti:Sa is used to generate fs pulses that are amplified in a regenerative amplifier ( Q -S Nd: YLF) but in no way a QML laser.
Finally the manuscript has certain critical errors . Figure 6a and 6b that must show two different pulses of 9ns and 2.1 ns, are the same.
In Conclusions, line 199, the pulse duration of QS regime is mentioned as 8.2 ns where it is 8.2 microsecons.
As a closing remark, I encourage the authors make an dynamical analysis of the Q- switching instabilities in this laser, that for sure it will be of great interest for the optical community. Unfortunately, at this stage I find no enough merits in this manuscript to be published.
Reviewer 2 Report
The author demonstrated a pulsed erbium-doped fiber laser based ontungsten trioxide as saturable absorber (SA). They reveal a pulse evolution process by adjusting the pump power and the polarization state in an erbium-doped fiber laser based on tungsten trioxide (WO3) SA. The WO3 based SA exhibited a large modulation depth of 20%. The paper is novel and includes some interesting and original results, may be interested to several researchers in this field. Therefore, I recommend to accept it for publication in Photonics, but with the following revisions.
1. What is the advantages of WO3? Any specific properties own by this material that make it special to be used as SA?
2. What is the insertion loss and polarization dependent loss (PDL) of the prepared SA?
3. How did the author determine that its size was 40nm? I suggest adding more representations, such as SEM, AFM, XRD.
4. Why do authors use optical deposition to prepare saturable absorbers? Why 25 mW for the pump laser What are the advantages over other methods, such as liquid phase stripping, CVD method.
5. The authors should supplement the doped fiber type and dispersion parameters. The manufacturer and address of the instrument used should be added as required by the publisher and magazine. What equipment was used to observe Fig. 1(a).
6. How about the damage intensity threshold of the WO3 SA?
7. The label in picture 4 is wrong. The author should explain the reason for the drop in pulse energy in Fig. 4(e).
8. Recently some SAs based on novel two-dimensional materials was proposed. (such as: 10.1016/j.optlastec.2022.107988, 10.1515/nanoph-2020-0267, 10.1515/nanoph-2019-0545, 10.1515/nanoph-2020-0116). Please compare their advantages and shortcomings.
Reviewer 3 Report
The authors demonstrated a pulsed fiber laser based on saturable absorber WO3 which can operate under three regimes of Q-switching, Q-switched mode-locking and mode-locking by simply tuning the pump laser power and the intra-cavity polarization state. This is a very good work.
However, there are some important information missing and minor errors
· What is the thickness of the deposited WO3 thin film?
· In your Q-switched mode-locked operation and the mode-locked operation, what are the polarizer angles for each state?
· The Figure 4 (c) should be ‘e’.
· Figure 6 (a) and (b) are the same, the pulse duration in (b) supposed to be 2.1 ns, but it shows 9.0 ns.
Finally, I suggest to accept after minor revision.